# Amyloid burden quantification depends on PET and MR image processing methodology

**Guilherme D. Kolinger**[1] *, **David Vállez García**[1], **Antoon T. M. Willemsen**[1], **Fransje E. Reesink**[2], **Bauke M. de Jong**[2], **Rudi A. J. O. Dierckx**[1], **Peter P. De Deyn**[2,3], **Ronald Boellaard**[1,4]

**1** Medical Imaging Center, University Medical Center Groningen, University of Groningen, Groningen, The Netherlands, **2** Department of Neurology, Alzheimer Research Centre, University Medical Center Groningen, University of Groningen, Groningen, The Netherlands, **3** Laboratory of Neurochemistry and Behaviour, Institute Born-Bunge, University of Antwerp, Antwerp, Belgium, **4** Department of Radiology and Nuclear Medicine, Amsterdam University Medical Center, VU Medical Center, Amsterdam, The Netherlands

* g.domingues.kolinger@umcg.nl

## Abstract

Quantification of amyloid load with positron emission tomography can be useful to assess Alzheimer's Disease *in-vivo*. However, quantification can be affected by the image processing methodology applied. This study's goal was to address how amyloid quantification is influenced by different semi-automatic image processing pipelines. Images were analysed in their *Native Space* and *Standard Space*; non-rigid spatial transformation methods based on maximum a posteriori approaches and tissue probability maps (TPM) for regularisation were explored. Furthermore, grey matter tissue segmentations were defined before and after spatial normalisation, and also using a population-based template. Five quantification metrics were analysed: two intensity-based, two volumetric-based, and one multi-parametric feature. Intensity-related metrics were not substantially affected by spatial normalisation and did not significantly depend on the grey matter segmentation method, with an impact similar to that expected from test-retest studies ($\leq$10%). Yet, volumetric and multi-parametric features were sensitive to the image processing methodology, with an overall variability up to 45%. Therefore, the analysis should be carried out in *Native Space* avoiding non-rigid spatial transformations. For analyses in *Standard Space*, spatial normalisation regularised by TPM is preferred. Volumetric-based measurements should be done in *Native Space*, while intensity-based metrics are more robust against differences in image processing pipelines.

**Data Availability Statement:** All relevant data are available at the Zenodo repository (https://doi.org/10.5281/zenodo.4353853).

## Introduction

Alzheimer's disease (AD) is a progressive neurodegenerative disease and the most common pathology leading to dementia in the elderly. Past [1] and current [2] guidelines provide means for the diagnosis of probable AD via clinical assessment of symptomatic patients. While proved pathological AD is demonstrated via *post-mortem* assessment [2,3], a research framework was proposed in which AD is defined by the pathological process underlying the patient's

**Funding:** GDK, DVG, and RB have received funding from the European Union's Horizon 2020 research and innovation programme under the Marie Skłodowska-Curie [grant agreement No 764458].

**Competing interests:** The authors have declared that no competing interests exist.

symptoms [4]. This definition is achieved with an *in-vivo* assessment of AD biomarkers, where positron emission tomography (PET) imaging plays an important role in the quantification of abnormal β-amyloid (Aβ) plaques depositions in the brain [5–7], one of the hallmarks of AD [8]. With the combination of PET/computed tomography (CT) with magnetic resonance imaging (MRI), it is possible to obtain detailed regional quantitative information about AD biomarkers in the brain. This hybrid imaging approach provides powerful means for the characterisation of early-stage AD and disease progression [9–11].

PET imaging can be assessed using fully quantitative methods applied to dynamic scans or semi-quantitatively with standardized uptake value ratios (SUVR) in static images that have shown to achieve high diagnostic accuracy, success for early disease detection, and differential diagnosis of AD [5,6,12–17]. While PET imaging provides quantitative data about the biological process under investigation, the anatomical volumes of interest cannot always easily be defined on the PET image due to its restricted spatial resolution and the fact that the PET signal is not always directly related to anatomy. On the other hand, MRI provides highly detailed anatomical images that can be aligned with PET images to perform a hybrid analysis, combining anatomical and functional data. Thus, it becomes possible to define anatomically derived volumes of interest on the MRI and perform PET quantification using these volumes.

Additionally, it is also possible to spatially normalise the images from the *Native Space* (defined by the MRI or the PET, and specific for each subject) to a *Standard Space*, commonly defined by the Montreal Neurological Institute (MNI) Space [18] that was created based on healthy young volunteers. With images in the *Standard Space*, it is possible to perform voxel-wise analysis to compare subjects, while regional analysis (based on anatomical volumes of interest) is possible in both *Native Space* and *Standard Space*. These spatial transformations are commonly calculated with the subject's anatomical MRI as reference. However, there is no consensus on the best approach for PET+MR image processing and only a few studies are exploring the impact of image processing approaches on PET quantification results [19–23]. Notably, there has been an initial exploration of a MR-less approach for spatial transformation of brain PET images into the MNI space using the low-dose CT acquired for attenuation correction during the PET/CT scanning [24]. However, in our study we focussed on pipelines for which both PET and MRI data are available, including the use of grey and white matter segmentations that are most commonly derived from T1-weighted MR images.

Non-rigid spatial transformations change the voxel volume, while the voxel intensities can be modulated. Some spatial transformation methods preserve the PET voxel concentrations and are expected to have less effects on voxel-wise analysis and metrics related to the activity concentration measured in the PET image. Other approaches preserve the total signal in the image and are expected to have less influence on measurements of total regional activity [25–28]. Furthermore, when carrying out analysis in *Standard Space* the tissue segmentation can be performed on the MR images before or after their non-rigid spatial transformation. This is not expected to strongly affect metrics, however, when pre-defined templates in *Standard Space* are used to delineate regions of interest on the PET images, a significant impact on metrics is expected. Unfortunately, the methodology applied to process the images and to define the volumes of interest is frequently underreported in literature and vary depending on the specific scientific question under investigation, rendering reproducibility and comparison between studies difficult.

The current study aimed to assess the influence of different image processing pipelines on quantification of amyloid burden with PET and to what extent various quantitative tracer uptake and volumetric measurements are affected by performing the feature extraction in *Native* or *Standard Space*, by using different spatial transformation methods, and by employing different grey matter tissue segmentation approaches.

## Materials and methods

### Subjects

Twenty-eight subjects underwent dynamic [$^{11}$C]PiB PET/CT scans for amyloid imaging, as well as a T1-Weighted 3D MRI scan. Twelve subjects were clinically diagnosed with AD following the National Institute on Aging and Alzheimer's Association (NIA-AA) criteria [2] and classified as β-amyloid positive based on visual inspection of the [$^{11}$C]PiB images, while sixteen subjects were healthy volunteers (HC) with no cognitive complaints and classified as β-amyloid negative. Healthy volunteers were recruited via advertisement on local newspapers and within the University Medical Center Groningen (UMCG). All subjects were considered competent (MMSE > 18) to give informed consent to participate in the study and a written informed consent was obtained. Subjects' demographics can be seen in Table 1. No statistically significant difference was found for age, weight, BMI, and injected activity between HC and AD subjects, while the MMSE difference was statistically significant (two-tailed t-test). The current cohort is a subset of a larger study at the memory clinic, Alzheimer Centre of the UMCG, Groningen, The Netherlands. Ethical approval was obtained from the Medical Ethical Committee of the UMCG (2014/320).

### Image acquisition and pre-processing

All PET/CT scans were performed with Siemens Biograph 40mCT or 64mCT scanners (Siemens Medical Solutions, USA) at the UMCG. Both systems were from the same vendor and generation; the acquisition and reconstruction protocols were harmonized, and the systems were (cross-)calibrated. No significant differences between the images provided by these two different scanners were found, as shown also in a previously published study using the same dataset [29]. Scans were performed under standard resting conditions with eyes closed. PET acquisition started at tracer injection and lasted at least 60 minutes. List-mode PET data was reconstructed with 3D OSEM (3i24s) including point spread function resolution modelling and time-of-flight. More details on the PET acquisition and reconstruction can be found elsewhere [30]. Using PMOD (version 3.8; PMOD Technologies LLC), the 40 to 60 minutes interval of the dynamic PET images was averaged to obtain a static PET image. MRI acquisitions were performed at the UMCG and nearby hospitals following local clinical routine 3D T1-weighted MRI acquisitions. MR images were re-sliced to a voxel size of $1.0 \times 1.0 \times 1.0$ mm$^3$, since some clinical scans had different voxel spacing ($0.98 \times 0.98 \times 1.0$ mm$^3$, $1.01 \times 1.01 \times 1.0$ mm$^3$, and $0.49 \times 0.49 \times 1.0$ mm$^3$).

### Native Space and Standard Space

In this study, the quantification of images was performed in two different reference spaces:

**Table 1. Subject demographics (mean ± standard deviation).**

|                          | AD      | HC      |
|--------------------------|---------|---------|
| Gender (M:F)             | 8:4     | 11:5    |
| Age (years)              | 64±8    | 69±5    |
| Weight (kg)              | 81±14   | 77±12   |
| BMI                      | 26±5    | 25±3    |
| MMSE                     | 24±3    | 30±1    |
| Minimum MMSE             | 19      | 28      |
| Injected activity (MBq)  | 352±75  | 386±46  |

1. *Native Space*: the T1-MRI image was used to define the reference space of each specific subject. In this *Native Space*, all the subject's brains are anatomically unique, and they all have their individual coordinate systems.

2. *Standard Space*: the T1-MRI of each subject was used to estimate the non-rigid spatial transformations needed to align and warp each subject-specific brain image into the MNI space. In this *Standard Space*, all the subject's brains are anatomically equivalent, and they share a single common coordinate system.

### Spatial transformations

Using Statistical Parametric Mapping version 12 (SPM12, Wellcome Trust Centre for Neuro-imaging, UK), static [$^{11}$C]PiB images were spatially aligned to the subjects' MRI (i.e. *Native Space*) using rigid transformations with a Normalised Mutual Information objective function and linear interpolation. Then, non-linear transformations were estimated to adjust each subject's MRI to the *Standard Space* using three different methods (details on the settings used can be found in Tables A and B in S1 File). Two methods are based on a maximum a posteriori (MAP) approach [25,26] that provides the same spatial transformation parameters while treating the voxel values differently: one method modulates the voxels to preserve the total amount of signal in the image (MAPa), while the other preserves the voxel concentrations (voxel intensities) of the original image (MAPc). The third spatial transformation method is a unified segmentation method (USM) that uses tissue probability maps (TPM) as deformable spatial priors for regularisation of the non-linear deformations and that preserves the voxel concentrations of the original image [27,28].

### Volumes of interest: Grey matter and cerebellum

For each subject, different volumes of interest (VOI) were created for the grey matter (GM) defined by a binary mask based on a 50% threshold of the GM TPM. The GM TPM was determined by either segmenting the subject's MRI (using the tissue segmentation tool in SPM12) or by using a GM template [31]. As such, three methods for defining GM tissue were used:

1. GM TPM in *Native Space* obtained from the tissue segmentation of the subject's MRI in its *Native Space* (MRn)

2. GM TPM in *Standard Space* obtained from the subject's spatially transformed MRI to *Standard Space* (MRs)

3. GM TPM from a template available in *Standard Space* (TEs)

After the definition of GM tissue, (inverse) spatial transformations were applied to move GM TPMs between *Native* and *Standard Space* using the three different methods (MAPa, MAPc, and USM) as described above. Thus, five sets of GM tissue maps were available: GM from MRn in both *Native* and *Standard Space*, GM from MRs in *Standard Space*, and GM from TEs in both *Native* and *Standard Space*. Notice that since spatial transformations were carried out with three different methods, three GM TPMs were available after moving to a different space. In this study, the cerebellum VOI was defined based on the Hammers atlas [31], which is available in the *Standard Space* (CERs) and can be transformed into *Native Space* using each of the three inverse spatial transformations.

### Image processing pipelines

The extraction of metrics was performed for each possible combination of reference space, spatial transformation method, and GM definition. Each of these combinations will be referred

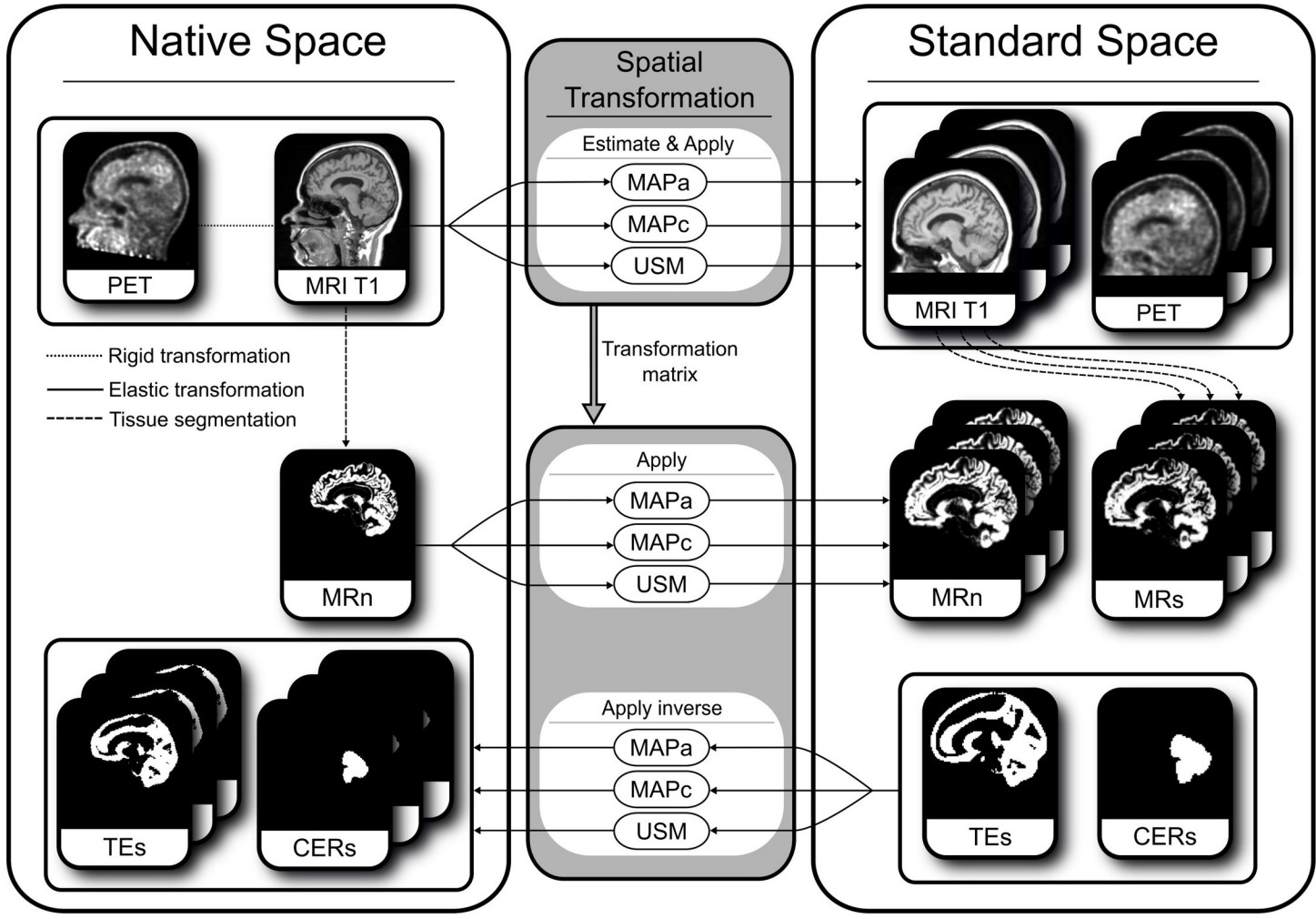

**Fig 1. Image processing pipelines scheme.** Scheme of the different image processing pipelines applied to the PET image, MRI, and grey matter (GM) tissue maps. The rigid transformation is shown by a dotted line, elastic transformations by continuous lines, and tissue segmentation by dashed lines. Images were transformed from *Native Space* into *Standard Space* using one of three possible spatial normalisation methods (upper and middle section). Spatial transformation matrices were calculated based on the MRI T1 image and these matrices were then applied to the other relevant images, while the inverse matrix was applied to transform the images from *Standard Space* to *Native Space*, for example when moving the GM derived from a standard template image or the cerebellum (CER) volume-of-interest (bottom section). Note that subscripts 's' and 'n' denote the space in which the tissue images were defined.

to as an *image processing pipeline*. Fig 1 shows a diagram of the different image processing pipelines assessed in the present study. Notice that spatial transformations can impact even the results obtained in the *Native Space* since the inverse transformations are applied to move the GM and the cerebellum VOI from the *Standard Space* into the *Native Space*. Two pipelines were considered as the main/reference pipelines: *Native Space* + MRn + USM and *Standard Space* + MRn + USM. In the first main pipeline, all images and pre-processing were performed in the patient *Native Space*. The second main pipeline involves a rigid transformation of the PET and MRI in *Native Space*, a GM segmentation using USM in *Native Space* followed by a non-rigid transformation of the PET, MRI, and GM images to *Standard Space*. These main pipelines represent the most commonly used ones for PET+MR data analysis in either reference space, as the tissue segmentation is performed before transformations, and a state-of-the-art method is used for spatial normalisation.

## Metrics

At each image processing pipeline, a set of SUVR images were generated by dividing each voxel value of the $[^{11}C]$PiB image by the average SUV obtained from the whole cerebellum's VOI. Then, the voxels with SUVR $\geq$ 1.5 were defined as PiB positive (A$\beta$+) [32–34] and corresponds to a high sensitivity for AD classification [35,36]. Finally, three sets of metrics were extracted from the GM VOI: intensity-based, volumetric, and multi-parametric.

Intensity metrics:

- Average SUVR of all voxels within the GM VOI (SUVR$_{mean}$)

- Average SUVR of A$\beta$+ voxels within the GM VOI (SUVR$_{meanA\beta+}$)

Volumetric measurements:

- Volume of A$\beta$+ voxels within the GM VOI (in mL)

- Amyloid Fractional Volume (AFV): the ratio between the A$\beta$+ volume and the GM VOI volume

Multi-parametric feature that combines intensity and volumetric metrics:

- Total Amyloid Burden (TAB) = SUVR$_{meanA\beta+}$ × A$\beta$+ volume

The extraction of these metrics was performed by in-house developed MATLAB (version R2018b) scripts based on SPM functions.

## Data analysis

Image processing pipelines were explored with boxplots and descriptive statistical analysis (mean and standard deviations are shown in the S1 File). The lower and upper whiskers of the boxplots extend to the 1.5×IQR (interquartile range), data beyond that are plotted individually. Additionally, an inferential statistical analysis was performed to explore the impact of the different pipelines on the metrics using a Generalised Estimating Equations (GEE) model [37–39]. The GEE model is known to achieve higher statistical power with small sample sizes than ANOVA [39] and does not require data normality [37]. For the present study, the GEE model was built with a linear scale response and an independent working correlation matrix. Space of analysis, spatial transformation method, and grey matter definition were included in the model as independent variables, as well as their interactions. The model was then applied independently to each metric (i.e. dependent variable). The overall results of the GEE model were reported with the mean difference, the 95% confidence interval, and the p-value in a table. These results are mentioned in the text were appropriate with the mean difference and p-value. A p-value of 0.05 was used as the threshold for considering statistical significance (Wald method, without correction for multiple comparisons in the pairwise comparisons). The results from the two main pipelines are presented in the main text in depth with the mean difference, 95% confidence interval, and p-value. The boxplot and descriptive statistical analyses were carried out with R (version 3.5.1, Rstudio version 1.1.456), while the GEE analyses were performed with SPSS (version 26.0.0.1, IBM Armonk, NY, USA).

## Results

In this section we will address the impact of using different reference spaces, spatial transformation methods, and GM definition methods separately for each of the amyloid burden uptake metrics, being intensity (SUVR), A$\beta$+ volume, amyloid fractional volume, and total amyloid burden. A collection of overall comparisons is presented in Table 2 and the two main

**Table 2. Overall comparisons of different pipelines.**

| Metric | Comparison | | Difference (%) | 95% Confidence interval (%) | p |
|---|---|---|---|---|---|
| SUVRmean | Reference Spaces | Standard—Native | 0.9 | 0.3–1.6 | < 0.01 |
| | Spatial Transformations | MAPc—MAPa | 0.6 | -0.4–1.6 | N.S. |
| | | USM—MAPa | 10.5 | 5.5–15.5 | < 0.001 |
| | | USM—MAPc | 9.9 | 5.2–14.6 | < 0.001 |
| | Grey Matter Definitions | MRs—MRn[†] | -0.2 | -0.3–0.0 | N.S. |
| | | TEs—MRn | 5.7 | -8.0–3.4 | < 0.001 |
| | | TEs—MRs | 5.6 | -8.0–3.3 | < 0.001 |
| SUVRmeanAβ+ | Reference Spaces | Standard—Native | -0.6 | -0.9 –-0.3 | < 0.001 |
| | Spatial Transformations | MAPc—MAPa | -0.0 | -0.5–0.5 | N.S. |
| | | USM—MAPa | 4.8 | 2.0–7.6 | < 0.01 |
| | | USM—MAPc | 4.8 | 2.2–7.4 | < 0.001 |
| | Grey Matter Definitions | MRs—MRn[†] | -0.1 | -0.2 –-0.0 | < 0.01 |
| | | TEs—MRn | 0.9 | 0.2–1.7 | < 0.05 |
| | | TEs—MRs | 1.3 | 0.4–2.1 | < 0.01 |
| Aβ+ Volume | Reference Spaces | Standard—Native | 13.8 | 5.9–21.7 | < 0.01 |
| | Spatial Transformations | MAPc—MAPa | 3.6 | -1.8–8.9 | N.S. |
| | | USM—MAPa | 36.0 | 16.5–55.4 | < 0.001 |
| | | USM—MAPc | 32.5 | 14.9–50.1 | < 0.001 |
| | Grey Matter Definitions | MRs—MRn[†] | 5.3 | 2.9–7.7 | < 0.001 |
| | | TEs—MRn | 30.8 | 20.9–40.8 | < 0.001 |
| | | TEs—MRs | 18.0 | 10.6–25.3 | < 0.001 |
| Amyloid Fractional Volume | Reference Spaces | Standard—Native | 0.3 | -2.6–3.1 | N.S. |
| | Spatial Transformations | MAPc—MAPa | 1.1 | -3.7–5.9 | N.S. |
| | | USM—MAPa | 35.6 | 16.6–54.6 | < 0.001 |
| | | USM—MAPc | 34.6 | 16.8–52.3 | < 0.001 |
| | Grey Matter Definitions | MRs—MRn[†] | -1.5 | -2.0 –-1.0 | < 0.001 |
| | | TEs—MRn | -9.9 | -18.1 –-1.7 | < 0.05 |
| | | TEs—MRs | -8.3 | -16.7–0.1 | N.S. |
| Total Amyloid Burden | Reference Spaces | Standard—Native | 13.4 | 5.6–21.2 | < 0.01 |
| | Spatial Transformations | MAPc—MAPa | 3.8 | -2.3–10.0 | N.S. |
| | | USM—MAPa | 45.0 | 21.0–68.9 | < 0.001 |
| | | USM—MAPc | 41.3 | 19.5–63.1 | < 0.001 |
| | Grey Matter Definitions | MRs—MRn[†] | 5.2 | 2.7–7.7 | < 0.001 |
| | | TEs—MRn | 30.4 | 19.9–40.8 | < 0.001 |
| | | TEs—MRs | 17.7 | 10.1–25.3 | < 0.001 |

[†]These comparisons were made in *Standard Space*. Statistical significance is presented in tiers: Non-significant (N.S.), p < 0.05, p < 0.01, and p < 0.001.

pipelines (*Native Space* + MRn + USM vs. *Standard Space* + MRn + USM) are compared in Table 3. At the end of this section, an overview summary of the main results is provided.

## Intensity metrics

**Comparing reference spaces.** The global effects from the GEE model (i.e., no interaction of the independent variables) found a small statistically significant overall difference of 0.92% (p = 0.007) in the $SUVR_{mean}$ values obtained in the different reference spaces, with *Native Space* having lower values than those in *Standard Space* (Table 2). Fig 2 and Table B in S1 File show $SUVR_{mean}$ distribution for different image processing pipelines and splits the subject

**Table 3. Comparing the main pipelines.**

| | SUVR$_{mean}$ | SUVR$_{meanA\beta+}$ | Volume A$\beta$+ | AFV | TAB |
|---|---|---|---|---|---|
| Difference (%) | 0.8 | -0.2 | 19.3 | 1.6 | 19.1 |
| 95% CI (%) | 0.5–1.0 | -0.3 –-0.0 | 11.5–27.1 | 0.5–2.7 | 11.0–27.2 |
| p | < 0.001 | < 0.05 | < 0.001 | < 0.01 | < 0.001 |

Comparing the two main pipelines: [*Standard Space* + MRn + USM]–[*Native Space* + MRn + USM].

groups by diagnosis. It was also possible to see that there was not a substantial difference in SUVR$_{mean}$ between the two main pipelines (0.8%, p<0.001, Table 3), that is, the change was less than what is expected from a repeatability analysis, even if it was statistically significant. Similarly, SUVR$_{meanA\beta+}$ was not substantially affected by analysis in different reference spaces

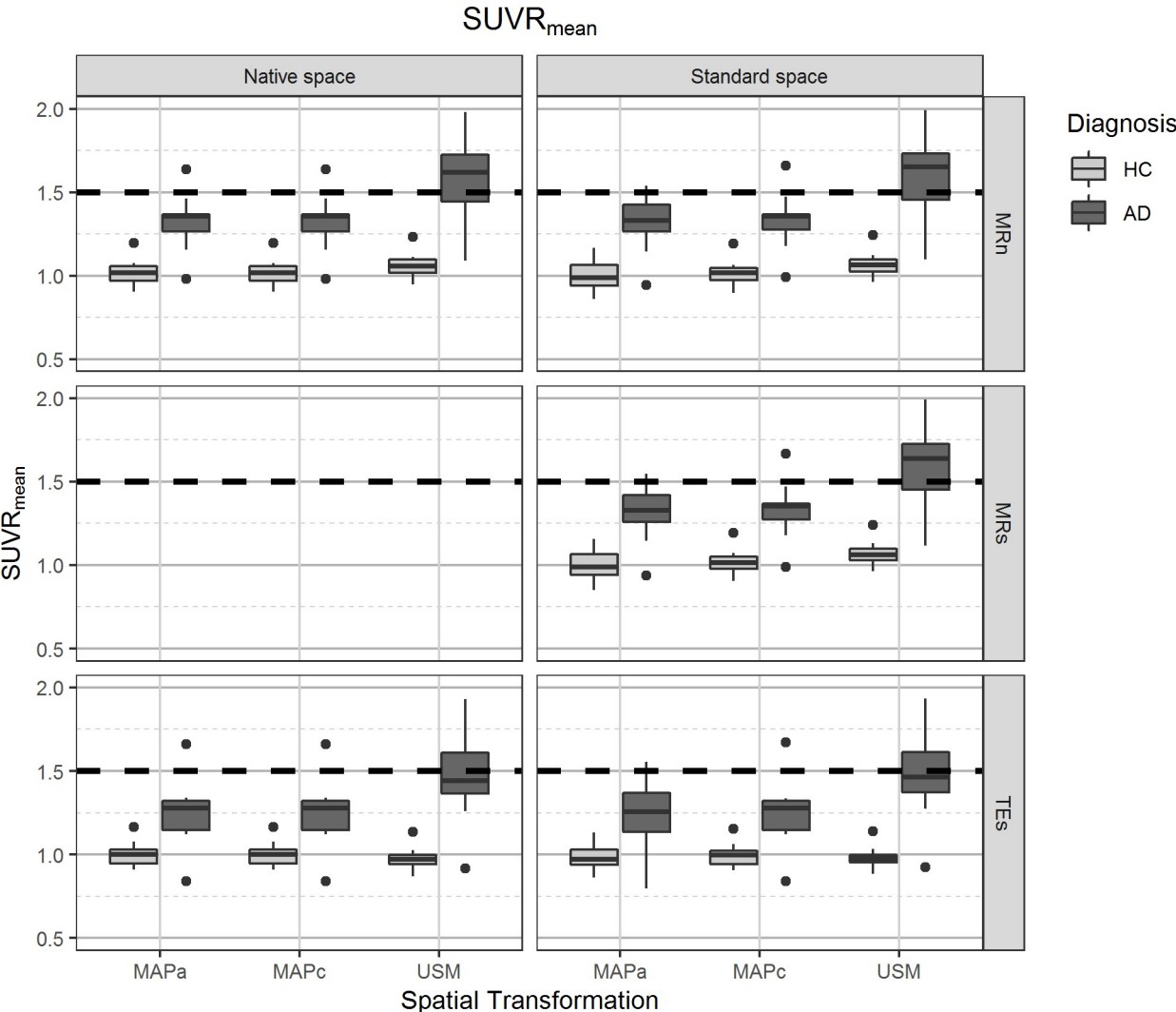

**Fig 2. Distribution of [$^{11}$C]PiB mean uptake ratio in grey matter tissue (SUVR$_{mean}$).** On the left panels, values from images analysed in the *Native Space*, on the right images that were spatially normalised to *Standard Space*. The different spatial normalisation methods are represented in sub-columns within each column, notice that spatial transformations for images in *Native Space* are required to transform template images from *Standard Space*. Rows show the different grey matter mask definitions. Diagnosis is indicated on the legend.

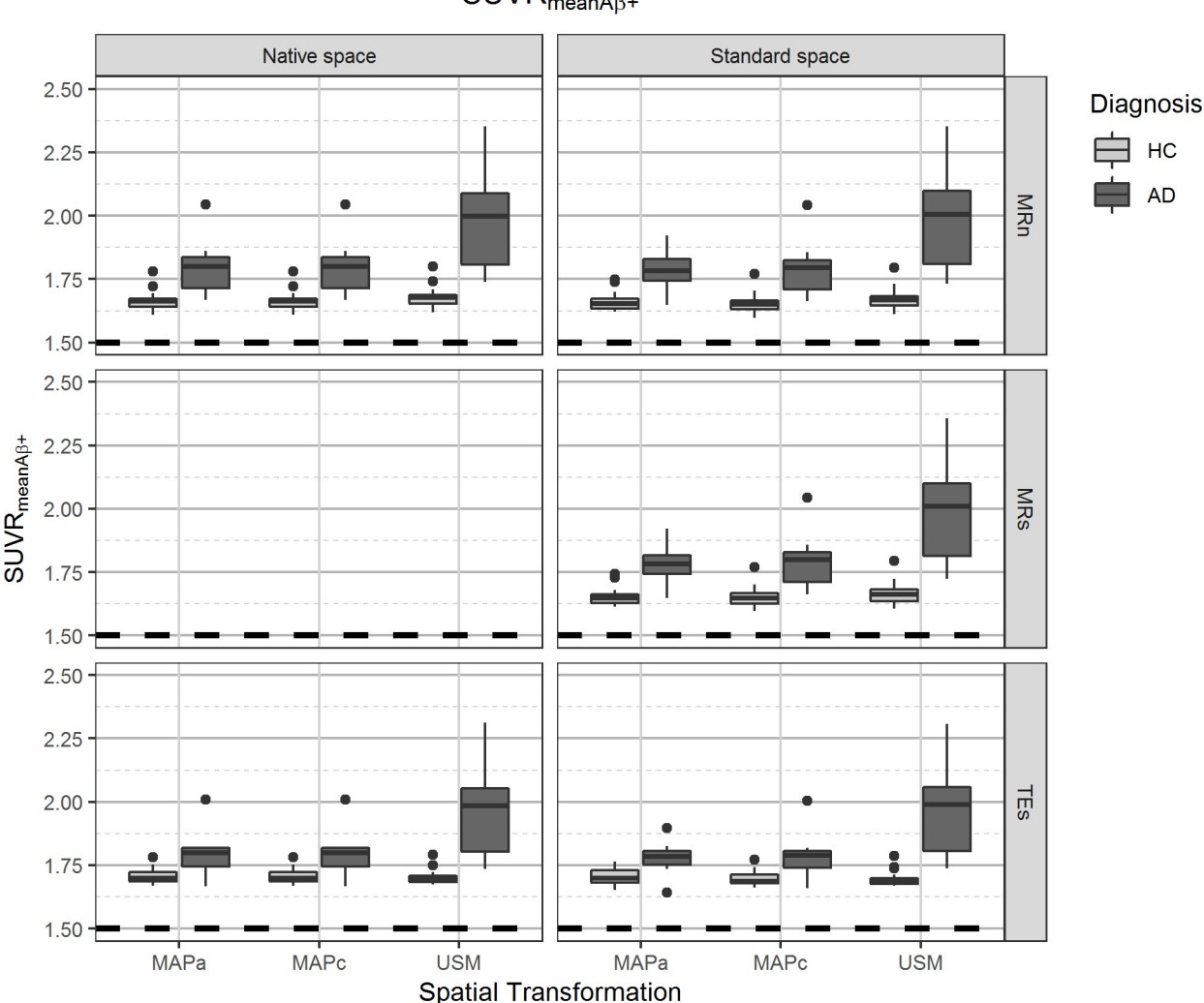

**Fig 3. Distribution of Aβ+ mean uptake in all grey matter tissue (SUVR$_{meanAβ+}$).** On the left panels, values from images analysed in the *Native Space*, on the right images that were spatially normalised to *Standard Space*. The different spatial normalisation methods are represented in sub-columns within each column, notice that spatial transformations for images in *Native Space* are required to transform template images from *Standard Space*. Rows show the different grey matter mask definitions. Diagnosis is indicated on the legend.

(Table 2). Fig 3 and Table B in S1 File show similar behaviour for SUVR$_{meanAβ+}$ as previously described for SUVR$_{mean}$. Difference between the two main pipelines was statistically significant, though with a very small effect size of -0.2% (p = 0.024, Table 3).

 **Comparing spatial transformation methods.** Overall, the SUVR$_{mean}$ values were only statistically and substantially affected by spatial transformation between the USM and the MAP methods, with USM yielding higher SUVR$_{mean}$ values (p<0.001, 10.5% and 9.9%, respectively for MAPa and MAPc). Fig 2 and Table B in S1 File show that there was a better separation between the subject populations when USM was used for spatial transformations, which also resulted in a larger spread of SUVR$_{mean}$ values. Likewise, there was not an overall statistically significant difference in using MAPa or MAPc for SUVR$_{meanAβ+}$ estimation. Meanwhile, USM produced slightly higher values than both MAP methods, however there was not a substantial impact in the metric (Table 2).

**Comparing GM definition methods.** Overall (global effects from the GEE model), slightly lower SUVR$_{mean}$ values were obtained using TEs than when the subject's MRI was used to define the grey matter tissue, either from MRn (-5.7%, p<0.001) or MRs (-5.6%, p<0.001). For analysis in *Standard Space*, there was no statistical difference on SUVR$_{mean}$ values between MRn and MRs. This can be seen in Table 2, Fig 3 and Table B in S1 File, which also show that all pipelines involving TEs make the separation between the two subject populations more difficult and set the average SUVR$_{mean}$ value below the Aβ positivity threshold (SUV = 1.5). Differences in overall SUVR$_{meanAβ+}$ values estimated from different GM tissue segmentations were not substantial, that is, smaller than that expected from a repeatability assessment (Table 2).

## Aβ+ volume

**Comparing reference spaces.** The overall mean Aβ+ volume obtained in *Standard Space* was 13.8% (p = 0.001) higher than that from *Native Space*. Fig 4 and Table 3 in S1 File show

## Aβ+ Volume

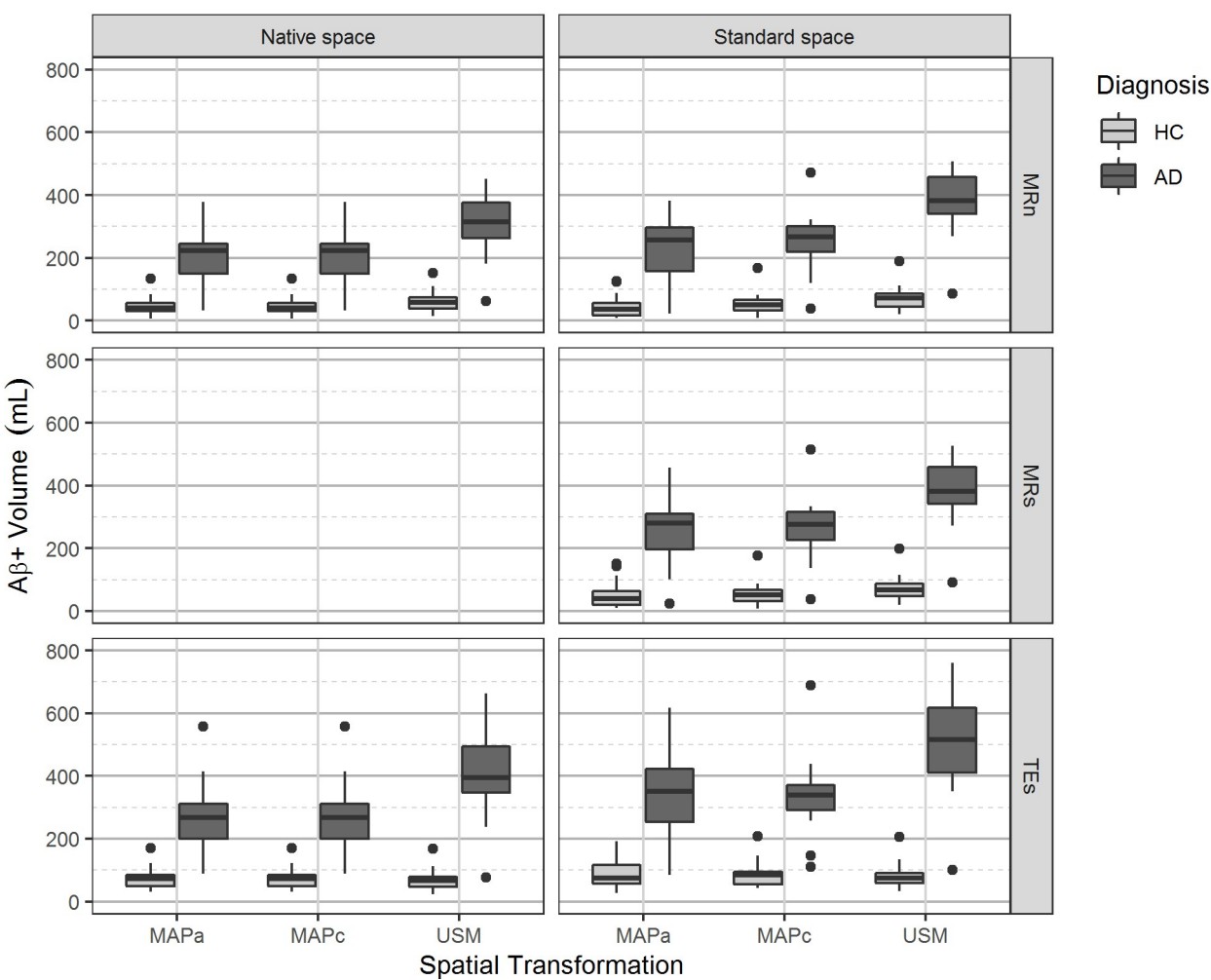

**Fig 4. Volume of high [$^{11}$C]PiB deposition region in grey matter (Aβ+ volume in GM).** On the left panels, values from images analysed in the *Native Space*, on the right images that were spatially normalised to *Standard Space*. The different spatial normalisation methods are represented in sub-columns within each column, notice that spatial transformations for images in *Native Space* are required to transform template images from *Standard Space*. Rows show the different grey matter mask definitions. Diagnosis is indicated on the legend.

that, as expected, AD subjects presented a higher Aβ+ volume than HC subjects for any image processing pipeline with analysis in either reference space, and indicates that Aβ+ volume measurements on AD subjects were more susceptible to spatial normalisation than HC subjects. Additionally, there was a significant difference between the two main image processing pipelines, with 19% (p<0.001) larger Aβ+ volumes in *Standard Space* than in *Native Space*.

**Comparing spatial transformation methods.** Overall larger Aβ+ volumes were obtained using with USM than with the MAPa and MAPc (36% and 33%, respectively; p<0.001 for both), and there was no significant difference between the two MAP methods themselves (Table 2). Spatial transformation with USM noticeably increased the separation between the two subject populations when compared with the MAP methods (Fig 4 and Table C in S1 File).

**Comparing GM definition methods.** Overall, Aβ+ volumes estimated with GM from TEs were 31% (p<0.001) larger than those that used MRn and 18% (p<0.001) larger than MRs. For analysis in *Standard Space*, there was no substantial difference between tissue segmentation from the T1-MRI before spatial transformation (MRn) or after (MRs).

## Amyloid fractional volume

**Comparing reference spaces.** Overall, the AFV values were robust and not statistically significantly affected by the estimation at *Native Space* or *Standard Space*. Fig 5 and Table C in S1 File show that the invariability of AFV against estimation on different reference spaces was not dependent on subject diagnosis. Comparing the two main image processing pipelines, only a 1.6% (p = 0.005) difference was found, with higher values in *Standard Space*.

**Comparing spatial transformation methods.** AFV was strongly affected by the spatial transformation method, with USM showing higher values than MAPa and MAPc methods (36% and 35%, respectively; p<0.001 for both). However, there was no statistically significant difference between the MAP methods (Table 2). Subject group separation was larger when using USM for spatial transformations when compared against the MAP methods (Fig 5 and Table C in S1 File).

**Comparing GM definition methods.** In general, AFV was only substantially and statistically significantly affected by GM definition when comparing TEs and MRn segmentation methods (-9.9%, p = 0.017, with lower values using TEs). In *Standard Space*, there was no substantial difference between GM segmentation before or after spatial normalisation of the MRI (Table 2).

## Total amyloid burden

**Comparing reference spaces.** TAB estimation was affected by analysis in different reference spaces, with 13% (p = 0.001) higher values in *Standard Space* than in *Native Space*. Fig 6 and Table D in S1 File show that TAB of AD subjects was slightly more impacted by the analysis in different spaces than that of HC subjects. Comparing the TAB estimated from the two main pipelines, 19% (p<0.001) higher values were found in *Native Space* than in *Standard Space*.

**Comparing spatial transformation methods.** There was no significant difference in TAB estimations between images processed with the MAP methods (Table 2). However, USM resulted in overall higher TAB values than MAPa (45%, p<0.001) and MAPc (41%, p<0.001). Fig 6 and Table D in S1 File indicate that there was a larger subject group separation when using USM than one of the MAP methods, regardless of reference space or GM tissue segmentation approach. However, USM did show a larger spread in TAB values for AD subjects than the MAP methods (TAB descriptive statistics for each pipeline available in the Table D in S1 File.

## Amyloid Fractional Volume

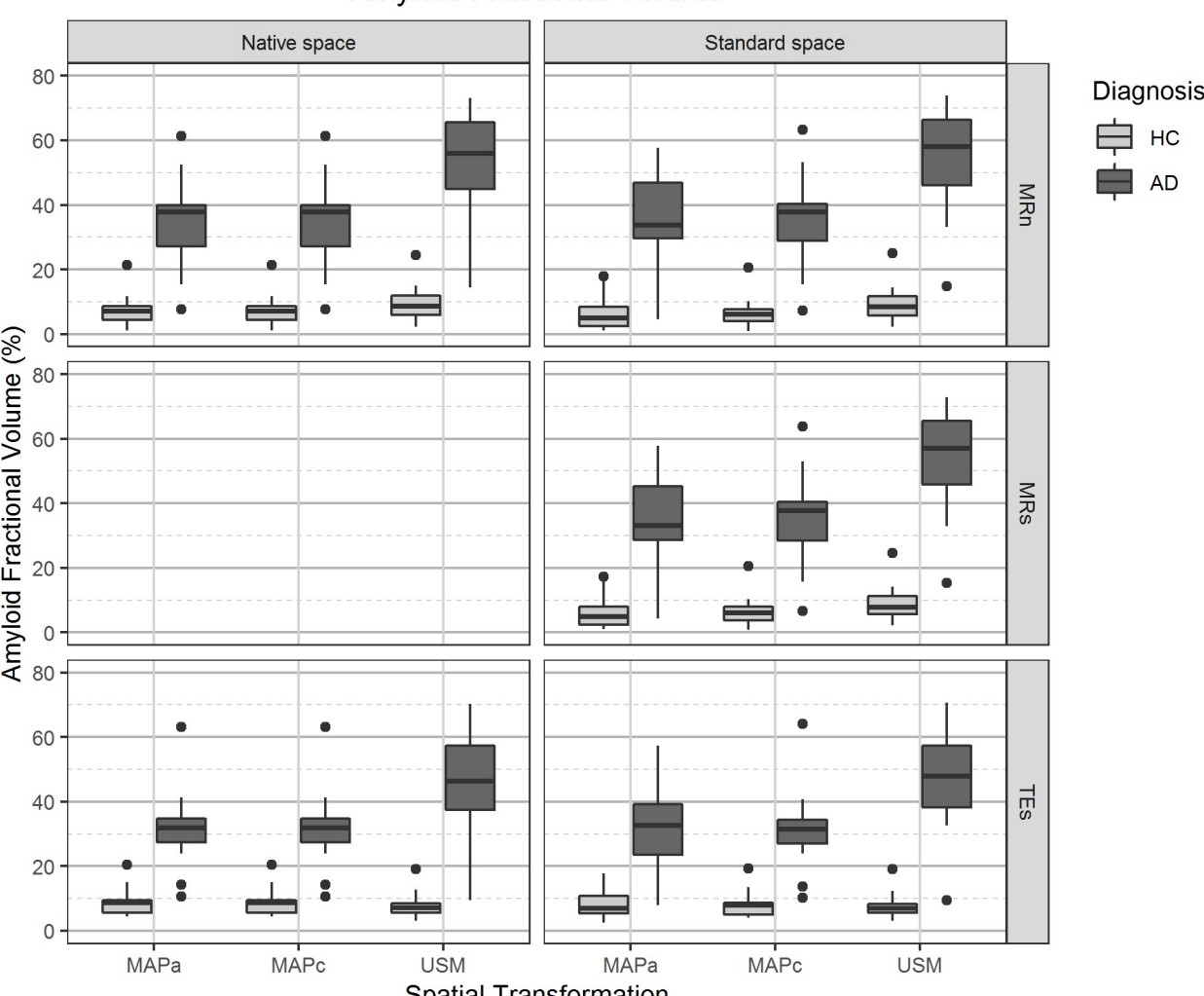

**Fig 5. Amyloid fractional volume (percentage of voxels within grey matter classified as Aβ+) values distribution.** On the left panels, values from images analysed in the *Native Space*, on the right images that were spatially normalised to *Standard Space*. The different spatial normalisation methods are represented in sub-columns within each column, notice that spatial transformations for images in *Native Space* are required to transform template images from *Standard Space*. Rows show the different grey matter mask definitions. Diagnosis is indicated on the legend.

**Comparing GM definition methods.** TAB values were highly dependent of GM tissue definition, with overall TAB values from TEs being higher than those from MRn (30%, p<0.001) and MRs (18%, p<0.001). For the analysis in *Standard Space*, there was a statistically significant difference on TAB estimated from MRn and MRs, though without a substantial effect size (Table 2). Fig 6 and Table D in S1 File show that there was a larger spread of TAB values from AD subjects when TEs was used in comparison to MRn and MRs.

In summary, it was found that the intensity metrics were not substantially affected by changes in the image processing pipeline, with a variability smaller than that expected from test-retest studies (~10%). Meanwhile, the volumetric metrics were heavily dependent on image processing pipeline (up to 36% overall differences). AFV showed more robustness to pipeline changes than Aβ+ volume, and only differences in spatial transformation methods led to substantial differences. TAB was also substantially impacted by image processing pipeline

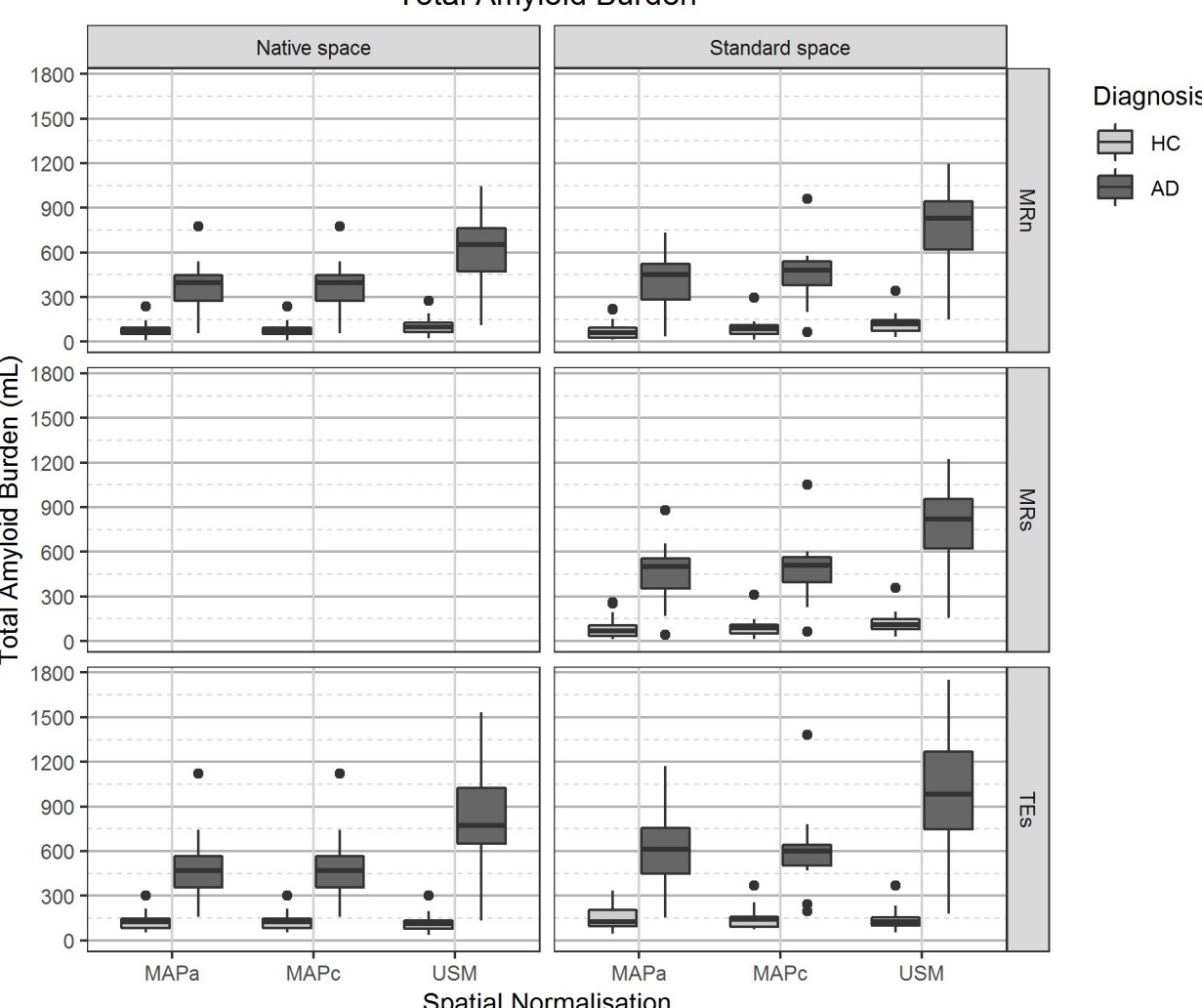

**Fig 6. Total amyloid burden value distribution.** On the left panels, values from images analysed in the *Native Space*, on the right images that were spatially normalised to *Standard Space*. The different spatial normalisation methods are represented in sub-columns within each column, notice that spatial transformations for images in *Native Space* are required to transform template images from *Standard Space*. Rows show the different grey matter mask definitions. Diagnosis is on the legend.

with overall variability reaching 45%. However, there was no substantial difference from MAPa to MAPc and MRn to MRs when estimating either Aβ+ volume or TAB. The two main pipelines resulted in statistically significantly different estimations for all metrics, but only Aβ + volume and TAB were substantially different, presenting 19% (p < 0.001) higher values in *Standard Space*.

## Discussion

In this study, the impact of different PET/CT+MR imaging processing pipelines on the quantification of intensity and volumetric-derived metrics was explored. These metrics were extracted after processing the images with different pipelines, that is, choosing a reference space (*Native Space* or *Standard Space*), a spatial transformation method (MAPa, MAPc, or USM) and a grey matter delineation, i.e. based on the subject's MRI (MRn and MRs) or using

a standard template (TEs). Overall, it was found that intensity metrics were not substantially affected (i.e., changes were less than the expected test-retest ($<10\%$) by the use of different image processing pipelines, while features using volumetric information were susceptible to the specific image processing pipeline used.

Quantification of intensity metrics, such as $SUVR_{mean}$ and $SUVR_{meanA\beta+}$, provided equivalent values when extracted in *Native Space* or *Standard Space*. A higher influence on the intensity metrics was observed when different spatial transformations were used, with overall differences reaching a difference of 10% when USM was used instead of one of the MAP methods. Furthermore, the definition of the grey matter tissue did not affect the intensity metrics in a substantial way (that is, less than what is expected from test-retest studies). As expected, there was no substantial difference between the two methods that used the subject's MRI for GM definition (MRn and MRs), though they did result in slightly different intensity metrics ($<6\%$) than a GM definition from a standard template (TEs). These (pairwise) differences were found to be statistically significant, however, the measured effect size was within the expected variability from [$^{11}$C]PiB test-retest studies [40–42] and, therefore, cannot be reliably measured. Comparing the two main image processing pipelines (*Native Space* + MRn + USM vs. *Standard Space* + MRn + USM), there were no clinically relevant differences in intensity metrics between them. Based on these results, it seems reasonable to conclude that $SUVR_{mean}$ and $SUVR_{meanA\beta+}$ are not notably affected by the image processing pipeline used, as the effect size observed was much smaller than the test-retest variability and will, therefore, have little impact on group studies.

The two volumetric measurements studied behaved differently. On one hand, Aβ+ volume was significantly affected by analysis in different reference spaces, showing that the images spatially transformed to *Standard Space* resulted in larger brain volumes than those in their *Native Space*. This can have an effect on the analysis of non-healthy brains in *Standard Space*, since the spatial transformations warp the brains into the MNI space [18], which was built with young healthy brains. Therefore, differences between healthy and non-healthy brains (that have smaller grey matter volumes due to atrophy, for example) are reduced on spatially transformed images. Hence, analysis of volumetric measurements should preferably be carried out in *Native Space* since spatial transformations may lead to bias, especially for AD subjects with strong atrophy. As such, studies reporting volumes or using the volumetric extend to determine disease stage should be interpreted with caution when comparing their results with data extracted in a different reference space. On the other hand, AFV values were not notably different from *Native Space* to *Standard Space* ($\leq 4\%$ difference, although statistically significant). Those volume-ratio metrics were more robust than the Aβ+ volume measurement when analysed in different reference spaces, and it is not surprising since the Aβ+ and GM VOI volumes will be scaled and spatially transformed similarly. It is important to notice, however, that localised brain atrophy information may be lost when spatially transformations into *Standard Space* are performed. Moreover, the use of spatial transformations based on USM method resulted in roughly 33–36% higher Aβ+ volumes and AFV values as compared to those obtained using the MAP methods, indicating that image transformation regularisation with TPM has a strong and significant impact on volumetric measurements, even when the ratio between volumes is considered. Note that USM is the most up-to-date spatial transformation method tested in the present study and previous literature suggest that it is an improvement over MAP methods [23]. Furthermore, the grey matter tissue definition heavily affected the estimation of the Aβ+ volume, with overall differences between pipelines on the range of 18–31% if a template GM was used instead of tissue segmentation directly from the subject's T1-MRI. This was expected as these GM templates are made such to guarantee the full inclusion of the actual GM regions of the subject and thereby overestimating the actual GM. AFV

values were somewhat less influenced by the grey matter definition, with around 9–10% difference between the GM estimation from the subject's MRI against a standard template. This reflects another consequence of the GM volume overestimation by these predefined templates, i.e. including voxels outside the actual subject's GM and that had a lower tracer uptake (SUVR < 1.5). Finally, comparing the two main image processing pipelines there was a significant and measurable difference in Aβ+ volume (19%, p<0.001), but the volumetric normalisation aspect of AFV made it robust between these pipelines (1.6%, p = 0.005). These results highlight the importance of harmonising the order in which the spatial transformations and the grey matter tissue segmentation are performed to obtain reproducible volumetric estimations for amyloid burden volumes.

Total amyloid burden combines $SUVR_{meanA\beta+}$ with Aβ+ volume and could potentially provide similar information to what Total Lesion Glycolysis gives in oncological [$^{18}$F]FDG studies, such as being a prognostic factor for disease progression, recurrence and death [43,44]. However, the variability due to changes in image processing pipeline observed for Aβ+ volume was reflected in TAB, which was significantly different between pipelines: overall significant changes from the analysis in different reference spaces was at 13%, due to different spatial transformations at 4–45%, and due to GM definitions on the range of 5–30%. By processing the images with the two main pipelines, a strong effect on TAB estimation between the two reference spaces (19%, p<0.001) was found. As such, the multi-parametric aspect of TAB did not compensate the variability from its volumetric component, but the other way around, it was amplified by its intensity component. Thus, it is uncertain to confirm the usefulness for an amyloid burden assessment with [$^{11}$C]PiB using a metric that is so susceptible to image processing methodology.

The effects of the image processing pipeline on the diagnostic discrimination power of each feature were not statistically evaluated since the HC and AD populations in the present study were chosen to have maximum separation and represent rather extreme cases, that is, no subjects with a doubtful classification or mild-cognitive impairment were included. As such, the statistical analysis was carried out pooling together both diagnoses to investigate the impact of different image processing pipelines on metrics regardless of clinical diagnosis. In any case, it was observed in the boxplots and descriptive statistics of each population that, in general, metrics from HC subjects were less affected by changes in image processing pipeline than measurements from AD subjects (Figs 2–6 and Supplementary Tables B–D in S1 File). This was expected as most features only included voxels with SUVR above 1.5 and AD subjects have more voxels that fit this criterion and a larger range of values above this threshold. A previous study [45] suggested that a lower SUVR threshold (SUVR≥1.2) for amyloid positivity and the use of the grey matter cerebellum as the reference tissue could improve sensitivity for diagnosis without impairing specificity. In the current study, the metrics were also assessed with a lower positivity threshold (SUVR≥1.2; see available data) and the general behaviour of the metrics was similar as the ones already presented (i.e. SUVR≥1.5). Therefore, changes in the SUVR threshold for amyloid burden positivity do not seem to overcome the variability observed in quantification from different image processing pipelines. That same study provides an answer about the use of different reference regions by stating that the use of the grey matter cerebellum as the reference region (instead of the whole cerebellum) increases the [$^{11}$C]PiB SUVR values by approximately 6%, thus simply rescaling the positivity threshold. Consequently, the impact of different image processing pipelines on metrics should remain similar to the one presented here if the GM cerebellum was considered as reference region.

The results obtained in the present study are in line with those published by Nørgaard et al. [19,20], where the influence of acquisition (PET scanner, acquisition protocol, reconstruction, and MR field strength), pre-processing steps (motion correction, co-registration, volume of

interest delineation strategy, and partial volume correction), and analysis (kinetic model and statistical analysis) on [$^{11}$C]DASB studies was explored. However, that study used a different tracer and metrics. The results of Nørgaard et al. and those observed in the present study demonstrate that different image processing methodologies can have a significant effect on PET quantification and support the idea that these findings are not restricted to a specific radiotracer or disease. Other studies [22,23] explored different methods for the spatial transformation of brain images and observed that methods which do not use TPM for regularisation of the transformations had worse performance than the regularised transformations, supporting the observed difference between the MAP and the USM methods in the current study. Furthermore, rigid transformation alignment errors can have an effect on SUVR quantification and lead to a bias related to clinical severity [46]. This may also be the case for non-linear transformations, possibly explaining why data from AD subjects were more affected by differences in image processing pipeline than those of HC subjects in the present study.

A possible limitation of this work is that the reference region and grey matter template masks were spatially transformed from *Standard Space* to *Native Space* using the inverse transformation calculated from the subjects' MRI. Therefore, MR-less pipelines which would benefit from template masks were not explored. However, it is uncommon to have MR-less dementia PET studies. Regional metrics were not evaluated, and target regions could be more sensitive to image processing than reference regions or global regions (as explored here).

In summary, the main finding of the present study is that intensity metrics (SUVRs) are rather robust against variations in image processing pipelines, and regional SUVR variability was within that expected from test-retest studies. In contrast, volumetric measurements are more sensitive to the image processing pipeline used. Based on the current results, regional-based studies should preferably be conducted in *Native Space* and tissue segmentation should be estimated from each subject's MRI when available, while standard VOIs (available in *Standard Space*) should be transformed into *Native Space*. Nevertheless, an inter-subject voxel-based analysis must be conducted in *Standard Space* and, for that matter, a standardised approach for spatial transformations must be employed, preferably using the Unified Segmentation Method (USM) since it uses tissue probability maps for transformations' regularisation. In either case, volumetric-based metrics should be avoided for studies in *Standard Space*. When comparing data and results with previous reports in the literature, interpretations should be made carefully since different conclusions could arise from methodological differences rather than from the biological process under investigation.

## Supporting information

**S1 File. Supplementary tables.** Include tables with additional information about parameters for spatial transformations, descriptive statistics of the metrics presented in the manuscript, and a table of abbreviations.
(DOCX)

## Author Contributions

**Conceptualization:** Guilherme D. Kolinger, David Vállez García, Ronald Boellaard.

**Data curation:** Guilherme D. Kolinger, David Vállez García.

**Formal analysis:** Guilherme D. Kolinger.

**Funding acquisition:** Peter P. De Deyn, Ronald Boellaard.

**Investigation:** Guilherme D. Kolinger, David Vállez García, Ronald Boellaard.

**Methodology:** Guilherme D. Kolinger, David Vállez García, Ronald Boellaard.

**Project administration:** Fransje E. Reesink, Bauke M. de Jong, Rudi A. J. O. Dierckx, Peter P. De Deyn, Ronald Boellaard.

**Resources:** Bauke M. de Jong, Rudi A. J. O. Dierckx.

**Software:** Guilherme D. Kolinger.

**Supervision:** David Vállez García, Ronald Boellaard.

**Validation:** Guilherme D. Kolinger.

**Writing – original draft:** Guilherme D. Kolinger, David Vállez García, Antoon T. M. Willemsen, Fransje E. Reesink, Bauke M. de Jong, Rudi A. J. O. Dierckx, Peter P. De Deyn, Ronald Boellaard.

**Writing – review & editing:** Guilherme D. Kolinger, David Vállez García, Antoon T. M. Willemsen, Fransje E. Reesink, Bauke M. de Jong, Rudi A. J. O. Dierckx, Peter P. De Deyn, Ronald Boellaard.

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
