## [Decision Letter · Decision Letter 0]

12 Jan 2021

PONE-D-20-39797

Amyloid burden quantification depends on PET and MR image processing methodology

PLOS ONE

Dear Dr. Kolinger,

Thank you for submitting your manuscript to PLOS ONE. After careful consideration, we feel that it has merit but does not fully meet PLOS ONE’s publication criteria as it currently stands. Therefore, we invite you to submit a revised version of the manuscript that addresses the points raised during the review process.

Please refine the paper following reviewers comments, in particular regard the methods adopted and results presentations.

We look forward to receiving your revised manuscript.

Kind regards,

Pierpaolo Alongi

Academic Editor

PLOS ONE

Journal Requirements:

2. Please amend your Methods section to state how (from where) the healthy volunteers were recruited.

3. Please provide additional details regarding participant consent.

In the ethics statement in the Methods and online submission information, please ensure that you have specified what type you obtained (for instance, written or verbal, and if verbal, how it was documented and witnessed).

If your study included minors, state whether you obtained consent from parents or guardians.

If the need for consent was waived by the ethics committee, please include this information.

Reviewers' comments:

Reviewer's Responses to Questions

**Comments to the Author**

1. Is the manuscript technically sound, and do the data support the conclusions?

Reviewer #1: Partly

Reviewer #2: Yes

2. Has the statistical analysis been performed appropriately and rigorously? 

Reviewer #1: Yes

Reviewer #2: Yes

3. Have the authors made all data underlying the findings in their manuscript fully available?

Reviewer #1: Yes

Reviewer #2: Yes

4. Is the manuscript presented in an intelligible fashion and written in standard English?

Reviewer #1: Yes

Reviewer #2: Yes

5. Review Comments to the Author

Reviewer #1: In this paper, the authors compare the results of PET amyloid quantification over different possible analysis pipelines. The paper nonetheless has several limitations. Both in the methods compared and in the metrics used for the comparisons.

The first doubt is in the "spatial transformations" paragraph. The authors used the "old normalize" toolbox twice, once modulating the transformation and once not. First "old normalize" toolbox for MR-based normalization is not standard anymore. Then, it's not clear why they modulated the transformation. Generally this is done only for VBM studies, not for PET ones. The authors should clarify what they expect from this choice. It's also possible to modulate the transformation with the more appropriate unified segmentation algorithm, but the authors decided not to do this. This should be justified. If USM is to be used anyway to achieve the segmentation, what is the point of running a knowingly sub-standard normalization? If authors want to compare different kind of normalization procedures they should probably focus on other (probably MR-less) methods. E.g.: a PET-templated based one (like Landau et al 2013, JNM) or maybe a CT-AC based one (Presotto et al 2018, NICL). Both pipelines are expected to be less precise than MR-based GM parcellation, but are more practical for clinical applications where MR scans might not be present. Therefore the comparison might be more useful.

Concerning the comparisons of GM Definition Methods, one expects that using a patient-specific segmentation vs a template should highlight a very strong dependence on the level of atrophy of each individual subject, as templates cannot exclude the presence of a higher percentage of CSF in the region of interest. As atrophy often correlates with amyloid load in AD pathologies, it would be better not to pool all patients together when comparing TE to other methods. It would be better studied as a function of atrophy probably (or amyloid load itself).

The authors should comment on the reasons for which an user might want to perform an analysis in native or in standard space.

Statistical analysis:

It's not explicit if analyses are performed using paired tests or not.

Specific major remarks:

line 136: "subject specific tissue probability maps". The TPMs in SPM are population based, not subject specific, therefore the sentence is unclear.

line 142: SPM tissue segmentation provides binary maps, not probability maps, unless I'm interpreting wrong what they authors mean.

line 144-147: Probably related to the previous point. If the GM maps are obtained from SPM USM, these aren't "probability maps" but a binary segmentation. So "TPM" shouldn't be used as it's a specific term meant to identify probability distributions.

line 146: The pipeline is not clear. Did the authors transform the MR to standard space and then applied once again USM? Or did they save the segmentation map in standard space directly from the segmentation algorithm? In the first case, which resampling voxel was used for the MR in native space transformation? If also the MR was resampled to 2 mm, I expect much poorer segmentation performance, as this pixel spacing is close to the GM thickness.

line 223: it is not clear how all analysis pipelines were pooled together and why.

line 227: It is claimed that no meaningful difference was found between the two main pipelines, but the difference is of the same order of the one identified as significant in line 222

line 270: The Ab volume was found to be 13.8% higher in standard space than in native space. This is almost by definition as the MNI space is defined to be bigger by about 10% than the average European brain. Therefore it is not clear how this quantity should be interpreted. Probably the fractional volume is more indicative then the absolute one.

Similarly, in many sections it is said that volumes (of different kinds) estimated with TE are larger than with MR segmentations. This is expected as GM segmentations are by definition thin strips, while templates necessarily cover wide areas. Therefore a comparison of "volume" between these two different kind of analyses is not really meaningful.

Line 359: it is said that intensity metrics do not depend in a statistically significant way from the analysis pipeline. But the results seem to differ. Also, as stated before, the fact that bias is not introduced by using TE instead of subject specific segmentation should be validated at least as a function of amiloyd load, due to the different amount of CSF present in the brain in different pathological states

Line 381: It is said that non-healthy brains have smaller volumes due to atrophy. I expect this to be the case only for gray matter volume, and not for the more-often used "total intracranial volume". The authors should clarify.

Line 383: What do authors mean by "exact volume measurements are required"?

Reviewer #2: This is a paper about the assessment of different processing methodology in the amyloid burden quantification in PET and MR images. The paper is well written and dense of statistical results. However, I’ll explain my few suggestions:

The manuscript is too long. The Authors should try to be more concise, trying to avoid any redundancy, if possible.

Authors should elicit why they did not consider the CT

Authors should consider using acronyms for Native Space and Standard Space

Lines 72 to 75: this period is too long, please rephrase in 2 different periods in a more clear form

Lines 227: “no meaningful” followed by a p<0.001 may be misleading, please rephrase in a more clear form

Please, use in all the figure “Native Space” instead of “Subject Space”

Add a list of the full abbreviation used at the end/beginning of the paper

If possible, add a key point section at the beginning of the paper since it is very extensive

6. PLOS authors have the option to publish the peer review history of their article (what does this mean?). If published, this will include your full peer review and any attached files.

Reviewer #1: No

Reviewer #2: **Yes: **Riccardo Laudicella

---

## [Author Response · Author response to Decision Letter 0]

12 Feb 2021

Reviewer #1

"In this paper, the authors compare the results of PET amyloid quantification over different possible analysis pipelines. The paper nonetheless has several limitations. Both in the methods compared and in the metrics used for the comparisons.

The first doubt is in the "spatial transformations" paragraph. The authors used the "old normalize" toolbox twice, once modulating the transformation and once not. First "old normalize" toolbox for MR-based normalization is not standard anymore. Then, it's not clear why they modulated the transformation. Generally this is done only for VBM studies, not for PET ones. The authors should clarify what they expect from this choice. It's also possible to modulate the transformation with the more appropriate unified segmentation algorithm, but the authors decided not to do this. This should be justified. If USM is to be used anyway to achieve the segmentation, what is the point of running a knowingly sub-standard normalization? If authors want to compare different kind of normalization procedures they should probably focus on other (probably MR-less) methods. E.g.: a PET-templated based one (like Landau et al 2013, JNM) or maybe a CT-AC based one (Presotto et al 2018, NICL). Both pipelines are expected to be less precise than MR-based GM parcellation, but are more practical for clinical applications where MR scans might not be present. Therefore the comparison might be more useful."

The ”old normalise” toolbox was explored as older pipelines stablished in several research groups may still be using SPM8 and therefore do not have access to the Unified Segmentation Method (USM). Widely available commercial packages such as PMOD still use SPM8 routines for spatial normalisation, while other freely available toolboxes such as PVELAB have only recently been updated to incorporate up-to-date spatial normalisation methods. Regarding the modulation of the intensity values, the transformation methods used mostly default SPM settings (with exception of bounding box and interpolation). As most literature failed to specify detailed settings for their transformations, the authors presumed that default settings were used. Since the goal was to replicate processing pipelines frequently used (and to keep the manuscript at a reasonable length), changes in the transformation parameters were not further explored. Furthermore, it is important to understand how future studies may be related to historical cohorts, highlighting the need for comparison of SPM12 with SPM8 spatial normalisation methods. Furthermore, quantitative brain PET studies are usually preceded by an MRI evaluation to assess if patients would benefit from a PET scan. As such, the authors focused on MR-pipelines and only explored the use of grey matter templates as an additional tool.

Regarding the use of the low-dose CT for spatial normalisations, Presotto et al 2018 present an interesting approach, which is now indicated in the introduction. However, in our study we focus on pipelines for which PET and MRI data are available and pipelines that make use of GM and WM segmentations from T1 weighted MR images. Yet we agree that it is worth mentioning the use of low-dose CT for normalization and that is why we have included a reference to this paper.

"Concerning the comparisons of GM Definition Methods, one expects that using a patient-specific segmentation vs a template should highlight a very strong dependence on the level of atrophy of each individual subject, as templates cannot exclude the presence of a higher percentage of CSF in the region of interest. As atrophy often correlates with amyloid load in AD pathologies, it would be better not to pool all patients together when comparing TE to other methods. It would be better studied as a function of atrophy probably (or amyloid load itself)."

The authors agree that a patient specific grey matter (GM) segmentation is expected to differ from a template-based tissue mask. This is especially important for subjects with severe atrophy since templates are built on healthy brains and cannot account for the GM atrophy on AD pathologies. Furthermore, grey matter templates typically overestimate GM volume to be sure that the actual subject’s GM region will be included fully in this standard volume of interest, further overestimating the GM volume in brains with atrophy. The authors have discussed this matter related to the MNI space in the main text’s Introduction and in the Discussion, however, for completeness, this was also mentioned for the GM tissue template in the revised version of the manuscript.

Fig 4 and Supplementary Table 3 show the influence of the image processing pipelines on the measured Aβ+ volume. It is possible to see that the Aβ+ volume of AD patients is more sensitive to changes in GM definition than HC subjects. Nevertheless, in the figure below (Fig R1) we can see that AD patients have a higher tracer uptake in grey matter while presenting a smaller GM volume than HC subjects, as is reflected on the figures and supplementary tables presented in the manuscript. This is not the case when a GM template (TEs) was used, showing that assessment of AD patients is more affected by GM definition method than HC subjects due to GM atrophy, as mentioned by the reviewer. Notice, however, that the manuscript does not discuss total grey matter volume but the amyloid positive volume within grey matter. 

[ Please see figure in the "Response to Reviewers" document ]

"The authors should comment on the reasons for which an user might want to perform an analysis in native or in standard space."

The Introduction section of the manuscript addresses this point: “With images in the Standard Space, it is possible to perform voxel-wise analysis to compare subjects, while regional analysis (based on anatomical volumes of interest) is possible in both Native Space and Standard Space.”

"Statistical analysis:

It's not explicit if analyses are performed using paired tests or not."

Statistical analysis performed pairwise comparisons, as is presented in the Data analysis section of the Methodology: “A p-value of 0.05 was used as the threshold for considering statistical significance (Wald method, without correction for multiple comparisons in the pairwise comparisons).”.

"Specific major remarks:

line 136: "subject specific tissue probability maps". The TPMs in SPM are population based, not subject specific, therefore the sentence is unclear."

Indeed, the tissue probability maps (TPM) in SPM are population-based. During the USM spatial normalisation, a non-linear deformation field is estimated to overlay these maps on the individual subject’s image. These warped-TPM into the subject’s images is what the authors meant by “subject specific tissue probability maps”. The authors agree that the text may be confusing, so it was updated to avoid misunderstandings.

"line 142: SPM tissue segmentation provides binary maps, not probability maps, unless I'm interpreting wrong what they authors mean.

line 144-147: Probably related to the previous point. If the GM maps are obtained from SPM USM, these aren't "probability maps" but a binary segmentation. So "TPM" shouldn't be used as it's a specific term meant to identify probability distributions."

Related to both these observations, the Segmentation module in SPM12 uses tissue probability maps to estimate each tissue class from an individual’s T1-weighted MRI. As such, TPMs that are specific for each subject and tissue class are generated as nifti files. In these maps, each voxel value represents the probability of that voxel belonging to such tissue class and the sum of all classes will be equal to 1. Below (Fig R2) you can see the TPM of grey matter for one of the subjects in the current study with colour representing the probability of each voxel belonging to this subject’s grey matter. The histogram (Fig R3) of GM probability values is also shown (after excluding the zeroes since all region outside the skull is zero and that would hide the distribution of non-zero voxels). This also justify why it was necessary to select a threshold (at 50%) for creating a grey matter binary mask from these TPMs.

[ Please see figures in the "Response to Reviewers" document ]

"line 146: The pipeline is not clear. Did the authors transform the MR to standard space and then applied once again USM? Or did they save the segmentation map in standard space directly from the segmentation algorithm? In the first case, which resampling voxel was used for the MR in native space transformation? If also the MR was resampled to 2 mm, I expect much poorer segmentation performance, as this pixel spacing is close to the GM thickness."

The reviewer is correct, the MR was spatially transformed to Standard Space and then the Segmentation module of SPM was used, so the USM was applied again. This simulated a workflow where all images are spatially normalised prior to tissue segmentation. This pipeline is indicated in Figure 1 by the top section (spatial transformation of the MRI T1) and then segmentation (dashed arrow) of the MRI in Standard Space (right section of the image).

Voxels were resampled to 2 mm with all spatial transformations, as indicated in the Supplementary Table 1, and, as the reviewer expected, the segmentation quality was worse when performed after the spatial normalisation of the MRI. The figure below (Fig R4) shows a zoom in a section of a TPM grey matter segmentation (both images in Standard Space). However, that did not lead to overall statistically significant differences in grey matter average SUVR (Table 2 of revised manuscript).

[ Please see figure in the "Response to Reviewers" document ]

"line 223: it is not clear how all analysis pipelines were pooled together and why."

The wording ‘pooling’ was poorly chosen. The Generalised Estimating Equations (GEE) model allows for assessment of a dependent variable as function of the interaction of independent variables. That is, the effects of an independent variable on a dependent variable considering the effects of other independent variable(s) simultaneously. GEE also permits for the overall assessment of the effects of an independent variable into a dependent one without considering interactions. By “pooling together analysis pipelines”, the authors referred to the global effects of an independent variable into a metric. As such, the GEE allowed for explanation of the variance from a global explanation (no interaction of the independent variables – what the authors originally mean by ‘pooling’ in this case) to subtle differences given by the interactions of the independent variables (for example, like the two main pipelines explored in the manuscript). Therefore, GEE performs this ‘top-down’ approach for assessment of independent variables effects on a dependent variable.

For clarity, the independent variables in the present study were reference space, spatial transformation method, and the grey matter definition method. The dependent variables were the metrics SUVRmean, SUVRmeanAβ+, Aβ+ volume, AFV, and TAB (each tested independently). The authors decided to focus on the overall effects (global explanation) of the dependent variables as a function of the independent variables to understand the “bigger picture” related to image processing pipeline effects on amyloid metrics. The main text was updated on the references of ‘pooling’ to avoid confusion and clarify to the reader what results are being presented.

"line 227: It is claimed that no meaningful difference was found between the two main pipelines, but the difference is of the same order of the one identified as significant in line 222"

What was meant by “meaningful” differences are those that are reliably measured on a PET scan, that is, if the difference was smaller than the test-retest of the metric then that difference was not considered meaningful even if statistically significant. However, the authors understand that the text can be confusing, so it was adjusted to reflect that changes smaller than test-retest (10%) are not considered substantial, regardless of statistical significance.

"line 270: The Ab volume was found to be 13.8% higher in standard space than in native space. This is almost by definition as the MNI space is defined to be bigger by about 10% than the average European brain. Therefore it is not clear how this quantity should be interpreted. Probably the fractional volume is more indicative then the absolute one."

The reviewer is correct, this result was not unexpected. In this paragraph, the authors are simply reporting the overall findings related to the measured Aβ+ volume in different reference spaces. It is also important to notice that Aβ+ volume also carries information about the tracer uptake as it counts all the voxels with SUVR ≥ 1.5 within the grey matter mask. Results on the Amyloid Fractional Volume (AFV) are reported afterwards. The Discussion section was updated to emphasise that this result was expected due to the definition of the MNI space itself. 

"Similarly, in many sections it is said that volumes (of different kinds) estimated with TE are larger than with MR segmentations. This is expected as GM segmentations are by definition thin strips, while templates necessarily cover wide areas. Therefore a comparison of "volume" between these two different kind of analyses is not really meaningful."

Larger volumes with template masks were indeed expected when compared against a subject specific tissue segmentation. The Discussion section was updated to include this point and avoid misinterpretation of the data. Notice, however, that amyloid fractional values were lower with template masks than with a proper grey matter segmentation. This was a consequence of the template including a bigger portion of voxels with lower uptake (SUVR < 1.5) than the tissue segmentation.

"Line 359: it is said that intensity metrics do not depend in a statistically significant way from the analysis pipeline. But the results seem to differ. Also, as stated before, the fact that bias is not introduced by using TE instead of subject specific segmentation should be validated at least as a function of amiloyd load, due to the different amount of CSF present in the brain in different pathological states"

Overall, the intensity metrics were either not statistically significantly affected by analysis pipeline or there was no substantial impact (i.e., varied less than their expected test-retest of 10%). The authors agree that the sentence is confusing and have rephrased it.

Regarding the use of a template grey matter mask (TEs), it did result in overall statistically significantly different SUVRmean than the tissue segmentation from the MRI T1 (6%, p<0.001, lower values with TEs than either MRn or MRs; Table 2of the revised manuscript). However, since this difference is lower than what would be expected from a test-retest study, the authors considered such impact not to be of clinical relevance since this difference cannot be reliably measured on a routine scan. When considering subject diagnosis, Fig 2 and Supplementary Table 2 (and Fig R1, included in this letter) show that SUVRmean of AD patients was more affected than that of HC when TE was used instead of a subject specific grey matter segmentation was performed.

"Line 381: It is said that non-healthy brains have smaller volumes due to atrophy. I expect this to be the case only for gray matter volume, and not for the more-often used "total intracranial volume". The authors should clarify."

Indeed, what the authors meant was the grey matter volume and the text was adjusted.

"Line 383: What do authors mean by "exact volume measurements are required"?"

What was meant is that spatial transformations may add a confounding factor to volumetric measurements as the methods use population-based templates for the spatial normalisation. This can be especially an issue for AD assessment due to atrophy. This could lead to bias on volumetric measurements, which would then not be exact. The manuscript was updated for clarification.

---

Reviewer #2

"This is a paper about the assessment of different processing methodology in the amyloid burden quantification in PET and MR images. The paper is well written and dense of statistical results. However, I’ll explain my few suggestions:

The manuscript is too long. The Authors should try to be more concise, trying to avoid any redundancy, if possible."

The authors attempted to present all the important results in a clear manner, with comparisons separated by sections and paragraphs. The authors included a new table (Table 2 of the revised manuscript) containing all the overall results from the GEE statistical analysis, and then were able to simplify the presentation of results within the text. This was also done for the comparison of the two main pipelines assessed (Table 3 of the revised manuscript). Hopefully, that makes the manuscript easier to read.

"Authors should elicit why they did not consider the CT"

In a research setting, quantitative brain PET studies are nearly always preceded by an MR scan. This practice is in place because of the high costs involved in a PET scan and the MRI scan is used to exclude unexpected abnormalities and thus to avoid that the PET scan will be acquired in vain. Due to the high anatomical resolution of this already available T1-weighted MRI, it is then used for tissue segmentation, volume of interest definition, and spatial transformations. In addition, the low-dose CT acquired in the PET/CT scan for attenuation correction does not have enough information for visual assessment or for a grey matter segmentation. In patient care, PET and MR images are also often both acquired and assessed visually. For these clinical studies, the low dose CT (from the PET/CT) could indeed also be used to align all image data. However, in our paper we preferred to focus on quantitative analysis pipelines for which PET and MR images are available and on pipelines that include patient specific GM segmentations.

"Authors should consider using acronyms for Native Space and Standard Space"

Although that could make some sections of the text shorter and less repetitive, the authors believe that it would make the manuscript more complicated to read as it would increase the number of acronyms being used. In fact, a table of abbreviations was added to the supplementary material to clarify the manuscript and adding more abbreviations or acronyms could make the paper harder to read for those less familiar with these topics. We hope that the reviewer is willing to agree with us.

"Lines 72 to 75: this period is too long, please rephrase in 2 different periods in a more clear form"

The text was updated to improve readability of the paragraph.

"Lines 227: “no meaningful” followed by a p<0.001 may be misleading, please rephrase in a more clear form"

By “no meaningful difference”, the authors meant that the differences are not substantial as they are smaller than the expected test-retest variability (~10%), even though a statistically significant result was found. The text was updated to avoid misinterpretation.

"Please, use in all the figure “Native Space” instead of “Subject Space”"

Figures were fixed to the correct label.

"Add a list of the full abbreviation used at the end/beginning of the paper"

The journal’s guidelines do not allow for an abbreviation list/table. However, the authors agree that this can help the reader to follow the manuscript and have created a supplementary table with the list of abbreviations.

"If possible, add a key point section at the beginning of the paper since it is very extensive"

The journal’s guidelines do not allow for a key point section. Nevertheless, the authors have added an overview of the main results at the end of the Results section (and a note to it in the beginning). Hopefully, this improves the readability and comprehension of the key points of the manuscript.

---

## [Editor Report · Decision Letter 1]

22 Feb 2021

Amyloid burden quantification depends on PET and MR image processing methodology

PONE-D-20-39797R1

Dear Dr. Kolinger,

We’re pleased to inform you that your manuscript has been judged scientifically suitable for publication and will be formally accepted for publication once it meets all outstanding technical requirements.

Kind regards,

Pierpaolo Alongi

Academic Editor

PLOS ONE
---

## [Editor Report · Acceptance letter]

26 Feb 2021

PONE-D-20-39797R1 

Amyloid burden quantification depends on PET and MR image processing methodology 

Dear Dr. Kolinger:

I'm pleased to inform you that your manuscript has been deemed suitable for publication in PLOS ONE. Congratulations! Your manuscript is now with our production department. 

Kind regards, 

on behalf of

Dr. Pierpaolo Alongi 

Academic Editor

PLOS ONE